# A pain-mediated neural signal induces relapse in murine autoimmune encephalomyelitis, a multiple sclerosis model

Yasunobu Arima[1,2†], Daisuke Kamimura[1,2†], Toru Atsumi[1,2], Masaya Harada[1,2], Tadafumi Kawamoto[3], Naoki Nishikawa[1,2,4], Andrea Stofkova[1,2], Takuto Ohki[1,2], Kotaro Higuchi[1,2], Yuji Morimoto[4], Peter Wieghofer[5], Yuka Okada[6], Yuki Mori[7], Saburo Sakoda[8], Shizuya Saika[6], Yoshichika Yoshioka[7], Issei Komuro[9], Toshihide Yamashita[10], Toshio Hirano[11], Marco Prinz[12], Masaaki Murakami[1,2]*

[1]Division of Molecular Neuroimmunology, Institute for Genetic Medicine, Graduate School of Medicine, Hokkaido University, Sapporo, Japan; [2]Laboratory of Developmental Immunology, Graduate School of Frontier Biosciences, Graduate School of Medicine, WPI Immunology Frontier Research Center, Osaka University, Osaka, Japan; [3]Department of Dentistry, Tsurumi University, Yokohama, Japan; [4]Department of Anesthesiology and Critical Care Medicine, Graduate School of Medicine, Hokkaido University, Sapporo, Japan; [5]Institute of Neuropathology, Faculty of Biology, University of Freiburg, Freiburg, Germany; [6]Department of Ophthalmology, Wakayama Medical University, Wakayama, Japan; [7]Laboratory of Biofunctional Imaging, WPI Immunology Frontier Research Center, Osaka University, Osaka, Japan; [8]Department of Neurology, National Hospital Organization Toneyama Hospital, Osaka, Japan; [9]Department of Cardiovascular Medicine, Graduate School of Medicine, University of Tokyo, Tokyo, Japan; [10]Laboratory of Molecular Neuroscience, Graduate School of Medicine, Graduate School of Frontier Biosciences, Osaka University, Osaka, Japan; [11]Osaka University, Osaka, Japan; [12]BIOSS Centre for Biological Signalling Studies, University of Freiburg, Freiburg, Germany

*For correspondence:
murakami@igm.hokudai.ac.jp

†These authors contributed equally to this work

Competing interests: The authors declare that no competing interests exist.

**Abstract** Although pain is a common symptom of various diseases and disorders, its contribution to disease pathogenesis is not well understood. Here we show using murine experimental autoimmune encephalomyelitis (EAE), a model for multiple sclerosis (MS), that pain induces EAE relapse. Mechanistic analysis showed that pain induction activates a sensory-sympathetic signal followed by a chemokine-mediated accumulation of MHC class II+CD11b+ cells that showed antigen-presentation activity at specific ventral vessels in the fifth lumbar cord of EAE-recovered mice. Following this accumulation, various immune cells including pathogenic CD4+ T cells recruited in the spinal cord in a manner dependent on a local chemokine inducer in endothelial cells, resulting in EAE relapse. Our results demonstrate that a pain-mediated neural signal can be transformed into an inflammation reaction at specific vessels to induce disease relapse, thus making this signal a potential therapeutic target.

## Introduction

Multiple sclerosis (MS) is genetically a T helper cell (CD4+ T cell)-mediated autoimmune disease (*International Multiple Sclerosis Genetics Consortium et al., 2011*) that is characterized by chronic

**eLife digest** Multiple sclerosis (or MS for short) is a disease in which the insulating covers of nerve cells in the brain and spinal cord become inflamed and damaged. Depending on which nerves are affected, this disease can cause a wide range of symptoms, ranging from numbness and muscle spasms to visual disturbances and chronic pain. Many other diseases and disorders also have pain as a symptom, but it is not well understood if pain itself can directly contribute to the development of disease.

Most people with MS will, initially, experience periods when their symptoms get worse (called 'relapses'), which are then followed by periods of improvement. Arima, Kamimura et al. investigated whether the sensation of pain itself could trigger a relapse in a mouse model of MS. The experiments showed that a painful sensation could trigger a relapse in the mice via the so-called 'gateway reflex'. This reflex describes the phenomenon whereby nerve impulses lead to the release of signaling molecules that cause the walls of nearby blood vessels to open and allow immune cells to move from the bloodstream to the central nervous system. This in turn stimulates the development of inflammation, which causes an imbalance in the affected sites of the central nervous system.

These findings demonstrate that pain itself triggers a signal—sent via nerve impulses followed by the release of signaling molecules—that can lead to a relapse; and suggest that interfering with this signal could potentially help to treat to protect against relapses in MS. Following on from this work, it will be important to confirm if the gateway reflex exists in humans, and whether it is linked to other diseases that don't involve the central nervous system.

inflammation in the central nervous system (CNS) and relapse in over two thirds of patients (*Steinman, 2009*). Such relapse is marked by inflammatory lesions consisting of various immune cells including CD4+ T cells, CD8+ T cells, B cells, and macrophages followed by a loss of neurological function in various regions of the CNS (*Steinman, 2014*). To understand the molecular mechanisms involved in the relapse, disease model animals including experimental autoimmune encephalomyelitis (EAE) mice have been designed to study the corresponding molecular mechanism (*Miller et al., 1995*; *Steinman, 2009*). However, a detailed explanation for the relapse is still lacking.

Pain induction is a symptom of MS (*Thompson et al., 2010*) and sometimes a determinant of MS activity (*Ehde et al., 2003*). Pain induction is commonly interpreted as a biomarker of inflammation, although inflammation can sometimes occur in the CNS without it (*Watson et al., 1991*). Yet a correlation between pain intensity and the pathogenesis of inflammation is poorly established and still debated (*Beiske et al., 2004*; *Kalia and O'Connor, 2005*; *Solaro et al., 2004*; *Ehde et al., 2006*). More specifically, it has not been demonstrated whether pain induction is involved in MS relapse.

One critical machinery for inflammation development is a local chemokine inducer in non-immune cells termed the inflammation amplifier (formerly IL-6 amplifier), which we originally discovered in several disease models including EAE (*Ogura et al., 2008*; *Hirano, 2010*). In this system, massive chemokine expression caused by the simultaneous stimulation of NFkB and STAT3 is followed by disruption of local homeostasis due to the local accumulation of various immune cells (*Murakami and Hirano, 2011*; *Murakami et al., 2011*; *Lee et al., 2012*, *2013*). Activation of the inflammation amplifier is associated with many human diseases and disorders such as autoimmune and neurodegenerative diseases including MS (*Murakami et al., 2013*; *Atsumi et al., 2014*).

Gateways for pathogenic CD4+ T cells in an EAE model have been established by activation of the inflammation amplifier via an anti-gravity-mediated regional neural signal in the fifth lumbar cord (L5) (*Arima et al., 2012*; *Mori et al., 2014*). Soleus muscle-mediated sensory-sympathetic neural signals enhance inflammation amplifier activation in L5 dorsal vessels via local norepinephrine expression derived from sympathetic neurons to create gates through which immune cells reach the CNS and cause inflammation there (*Arima et al., 2012*). These gates can be artificially manipulated via the activation of local neural signals by electrically stimulating muscles, resulting in an accumulation of immune cells in the intended regions (*Arima et al., 2012*). Gating blood vessels by regional neural stimulations, a phenomenon termed the gateway reflex, is an example of the neural signaling-mediated regulation of inflammation (*Tracey, 2012*; *Sabharwal et al., 2014*) and has potential therapeutic value not only in preventing autoimmunity, but also in augmenting the effects of

immunotherapies against infections and cancers (*Ogura et al., 2013*; *Atsumi et al., 2014*). Because it is known that neural activations are induced by various situations including pain (*Delmas et al., 2011*; *Palazzo et al., 2012*; *LaMotte et al., 2014*), we considered whether neural signals mediated by pain induction plays a role in the pathogenesis of EAE relapse via the gating of certain blood vessels.

In the present paper, we induced a pain stimulus in EAE-recovered mice that had minimal EAE symptoms. The result was EAE relapse. Mechanistic analysis demonstrated that a neural signal via sensory-mediated sympathetic activation leads to an accumulation of MHC class II+CD11b+ cells at ventral vessels of the L5 cord (L5 ventral vessels) in a manner dependent on CX3CL1 chemokine expression followed by an accumulation of various immune cells including pathogenic CD4+ T cells. These results demonstrate that pain-mediated neural signals risk reactivating inflammation via the accumulation of immune cells at specific sites and therefore may offer a new therapeutic target for relapse in chronic inflammatory diseases like MS.

## Results

### Pain induction develops the relapse of EAE

We have previously employed a passive transfer method for EAE induction (*Ogura et al., 2008*; *Arima et al., 2012*). We isolated myelin oligodendrocyte glycoprotein (MOG) specific CD4+ T cells from C57BL/6 mice having an immunization with MOG plus CFA and stimulated the resulting CD4+ T cells with the MOG peptide and antigen presenting cells in vitro. The activated MOG-specific CD4+ T cells (pathogenic CD4+ T cells), which include Th1 and Th17 cells, were intravenously injected into wild type C57BL/6 mice. EAE was induced within 1 week and disappeared at about 2–3 weeks after the transfer (*Figure 1A*). We refer to this disappearance as the remission phase and these mice as EAE-recovered. In contrast, we found no relapse even more than 300 days after the pathogenic T cell transfer (*Figure 1A*). Consistent with these results, it is known that the lack of natural relapses in the C57BL/6 mouse strain. Thus, our EAE model develops a transient clinical symptom upon the transfusion of pathogenic CD4+ T cells that is followed by the remission phase.

To show whether pain induction is involved in the development of EAE, we ligated the middle branch of the trigeminal nerves, which is composed of only sensory neurons (*Thygesen et al., 2009*). Although this pain induction did not accelerate disease development, particularly during the onset period (until 10 days after the pathogenic T cell transfer), the pain persisted and maintained symptoms at a high level from 15 days post T cell transfer compared with sham mice (*Figure 1B*). We found that pain induction itself (without pathogenic T cell transfer) did not induce the development of EAE (*Figure 1B*) and that pain induced in mice with pathogenic T cells gradually caused a low clinical score over 20 days after the transfer (*Figure 1B*). We found negligible cfos expression in neurons of the somatosensory area, which is a key area involved in pain sensation (*Li et al., 2010*), in the brain of EAE-recovered mice and negligible pain sensation according to the von Frey test of the extremity, trunk of the body, face, head (*Figure 1C* and data not shown), suggesting that remittent mice have minimal pain. Then we investigated whether pain induction was involved in the relapse of EAE. We induced pain in EAE-recovered mice that had developed transient EAE. Pain induction resulted in a relapse of EAE (*Figure 1D*).

Finally, it is important to investigate whether pain induction itself was involved in the EAE relapse. Pain medicines such as Gabapentin and/or Pregabalin, which reduced cfos expression in neurons of the somatosensory area or von Frey test values after pain-induction (data not shown), suppressed the development of EAE relapse (*Figure 1E*). On the other hand, capsaicin, a painful agent, injected into the whiskers or forefeet caused EAE relapse (*Figure 1F*). A relapsing-remittent model of SJL/PLP mice was also suppressed by pain medicines (*Figure 1G*). Thus, pain induction developed relapse in EAE models.

### Pain-mediated sensory activation is involved in EAE relapse

We further investigated whether the sensory pathway is involved in the pain-induced EAE relapse. Activated neurons, as defined by NeuN and cfos expressions, expressed the sensory markers TRPV1 and/or Nav1.8 in trigeminal ganglions. We found negligible cfos expression, however, in the subtypes of these markers in the absence of ligation in the remittent phase (*Figure 2A*). Immunohistochemical experiments using serial sections showed that cfos+-activated neurons also expressed TRPV1 and/or Nav1.8 (*Figure 2B*). Treatment with a Nav1.8 blocker, A803467, suppressed the development of the

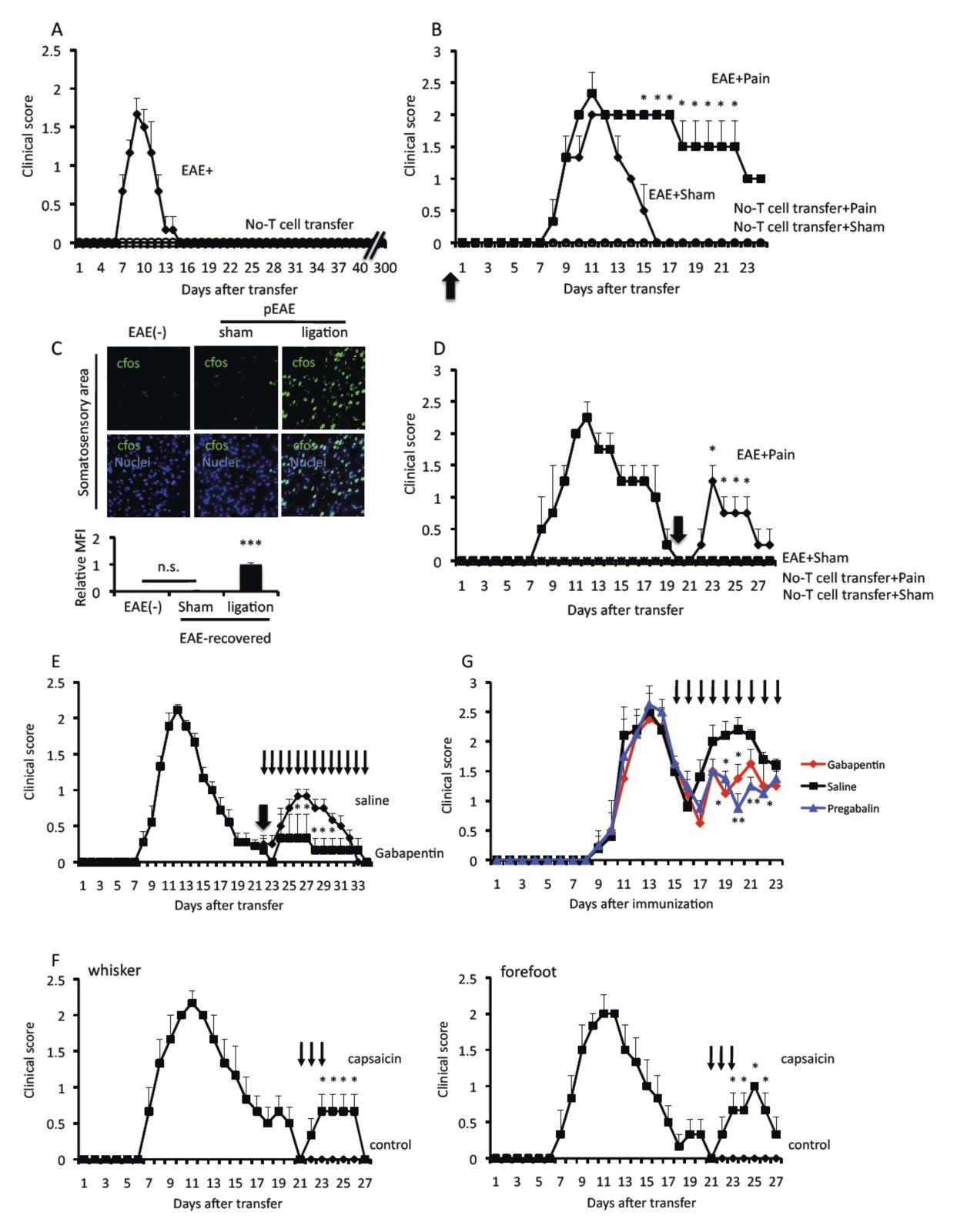

**Figure 1.** Pain induction causes EAE relapse. Pathogenic CD4+ T cells isolated from EAE mice were intravenously transferred into wild type C57BL/6 mice. (**A**) EAE development without pain induction (n = 8–10 per group). (**B**) EAE development with or without pain induction on the same day of T cell transfer (day 0, thick arrow) (n = 3–5 per group). (**C**) Immunohistochemical staining for cfos in the somatosensory area of wild type mice and EAE-recovered mice and EAE-recovered mice with pain induction (n = 2–3 per group) (top). Quantification of the histological analysis of the ACC (bottom). (**D**) EAE

*Figure 1. continued on next page*

*Figure 1. Continued*

development with or without pain induction 20 days (thick arrow) after T cell transfer in EAE-recovered mice (n = 3–5 per group). (**E**) EAE development in an EAE relapse induced by pain model in the presence or absence of Gabapentin (day 22–34, thin arrows). Pain was induced 22 days after the T cell transfer in EAE-recovered mice (thick arrow) (n = 3–5 per group). (**F**) EAE development with or without capsaicin treatment (day 21–23, thin arrows) in the whiskers or forefeet (n = 3–5 per group). (**G**) EAE development in a relapsing-remittent model in the presence Gabapentin (red), Pregabalin (blue), or saline (black) everyday from day 15 after immunization (thin arrows) (n = 4–5 per group). Mean scores ± SEM are shown. *, p < 0.05, **, p < 0.01, ***, p < 0.001, n.s., not significant. Experiments were performed at least 3 times; representative data are shown.

The following figure supplements are available for figure 1:

**Figure supplement 1**. Ligation of the middle branch of trigeminal neurons increased von Frey test values.

**Figure supplement 2**. Mice treated with capsaicin enhanced the accumulation of MHC class II+CD11b+ cells and cfos expression in neurons of the somatosensory area without pain induction.

EAE relapse (*Figure 2C*), and TRPV1-deficient hosts showed suppressed relapse (*Figure 2D*). We also found that animals treated with A803467 and TRPV1-deficient hosts showed reduced cfos expression in neurons of the somatosensory area or low von Frey values after pain-induction (*Figure 2—figure supplement 1*, right panels and data not shown). Moreover, ligation of the facial nerves, which mainly contain motor nerves and the vagus nerves, which mainly contain efferent fibers such as parasympathetic and motor nerves to various viscera induced negligible development of EAE relapse (*Figure 2—figure supplement 2* and data not shown). Therefore, we concluded that the activation of sensory pathways are involved in the pain-induced relapse.

## Pain-mediated sympathetic activation is involved in EAE relapse

We then investigated whether sympathetic activation is also involved in EAE relapse after pain induction. We first investigated blood flow speed, which is controlled by autonomic neurons including sympathetic ones, in the dorsal vessels of L5 and L1, the bottoms of the forefeet, the hindlimb, femoral vessels, and brain surface vessels in pain-induced mice. Blood flow speed was faster in pain-induced than sham-operated mice (*Figure 3A*). These results suggested that sympathetic activation affects blood vessels in the CNS after pain induction. Consistent with these results, neurons of the somatosensory area, particularly the anterior cingulate cortex (ACC), where sensory nerves are localized with autonomic nerves including sympathetic ones (*Ikemoto et al., 1999*; *Critchley, 2005*), were activated according to cfos expression (*Figure 1C*). Moreover, sympathetic ganglions were activated after pain induction, with the activation status in L5 being higher than in other cords (*Figure 3B*). Furthermore, treatment with atenolol, an inhibitor of the norepinephrine β1 receptor, significantly suppressed the relapse development (*Figure 3C*), and treatment with 6-OHDA, which induces sympathectomy, suppressed the EAE relapse (*Figure 3D*), while mice treated with atenolol and 6-OHDA did not suppress cfos expression in neurons of the somatosensory area after pain induction (*Figure 3—figure supplement 1*). Moreover, EAE mice under immobilization stress or forced swimming did not show EAE relapse, although these conditions did increase serum corticosterone, norepinephrine, and epinephrine similarly to pain induction (*Figure 3E,F*). Therefore, these data suggest specific sympathetic pathways triggered by pain induction, but not pain-mediated systemic hormonal responses and/or stress-mediated events, play a role in the development of EAE relapse.

## A pain-mediated neural pathway accumulates MHC class II+CD11b+ cells at L5 ventral vessels via sympathetic-mediated chemokine expression in EAE-recovered mice

We hypothesized that inflammation could be induced in the L5 cord after pain induction, because the disease symptoms of the EAE relapse after pain induction were similar to the symptoms developed by primary EAE whose inflammation was initially developed in the L5 cord after pathogenic CD4+ T cell transfer (*Arima et al., 2012*). Many cell populations increased in the L5 cord after pain induction (*Figure 4A*), especially those of CD11b+ cells and particularly MHC class II+ ones, in EAE-recovered

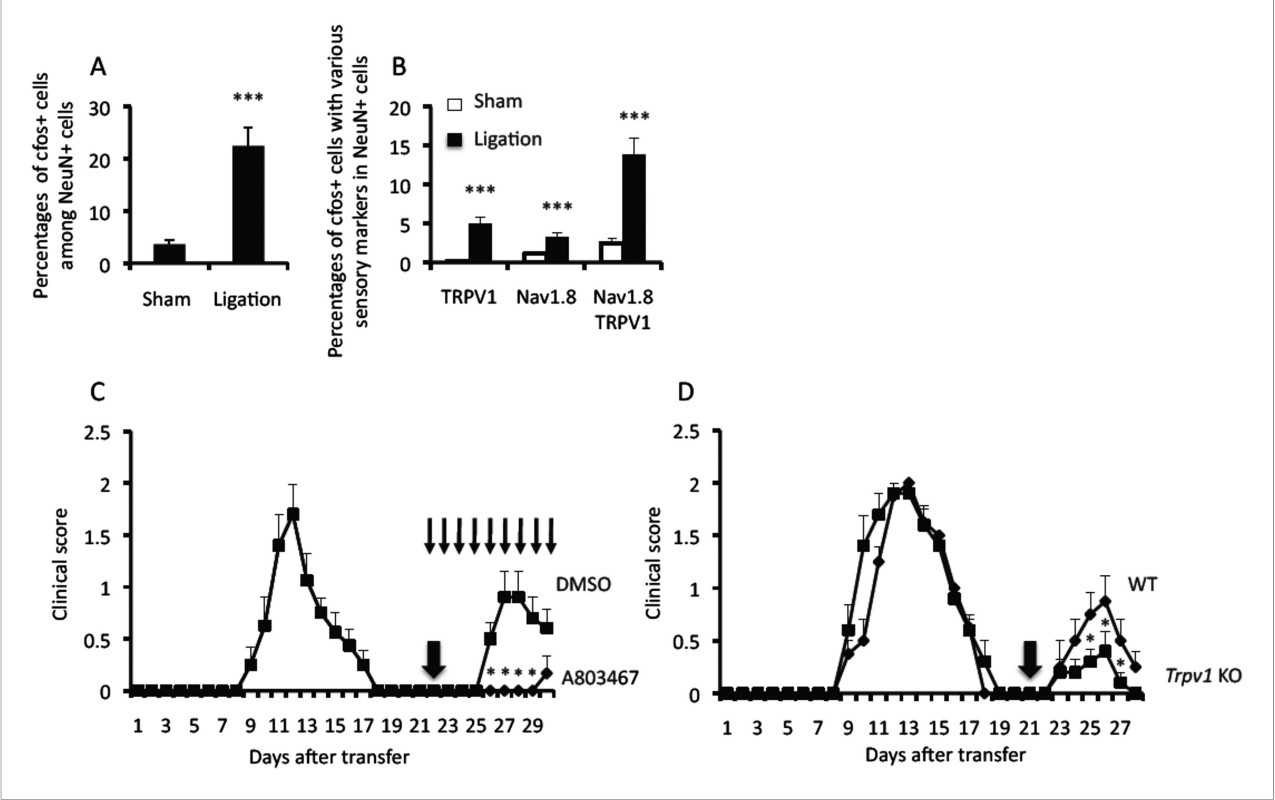

**Figure 2**. Sensory activation is involved in EAE relapse. (**A**) Percentages of cfos+ cells among NeuN+ cells in the presence or absence of ligation in the trigeminal ganglions of wild type mice (n = 2–3 per group). (**B**) Percentages of cfos+ cells with sensory markers TRPV1 and/or Nav1.8 in the presence or absence of ligation of the trigeminal ganglions of wild type mice (n = 2–3 per group). (**C**) EAE development with or without A803467 treatment (day 22–30, thin arrows) in the presence of pain induction 22 days (thick arrow) after T cell transfer in EAE-recovered mice (n = 4–5 per group). (**D**) EAE development in TRPV1 deficient mice in the presence of pain induction 21 days (arrow) after T cell transfer in EAE-recovered mice (n = 4–5). Mean scores ± SEM are shown. *, p < 0.05, ***, p < 0.001, n.s., not significant. Experiments were performed at least 3 times; representative data are shown.

The following figure supplements are available for figure 2:

**Figure supplement 1**. Mice treated with A803467 and TRPV1-deficient hosts suppressed the accumulation of MHC class II+CD11b+ cells and cfos expression in neurons of the somatosensory area after pain induction.

**Figure supplement 2**. Mice treated with ligation via vagus nerves did not induce the accumulation of MHC class II+CD11b+ cells.

mice. Furthermore, the accumulated CD4+ T cells in early phase (day 1–2) after pain induction were mainly Th17 cells, while in later time points (day 5 and later) after pain induction, we found not only other helper subset in L5 cord and lymphocytes accumulations in several regions of brain just like primary EAE responses (data not shown).

We performed parabiosis experiments using CD45.1 and CD45.2 hosts to investigate whether MHC class II+CD11b+ cells in the L5 of EAE-recovered mice originated from resident microglial cells or peripheral monocytes. We found similar percentages of CD45.1+ cells and peripheral derived CD45.2+ cells in MHC class II+CD11b+ cells in the L5 cord of EAE remittent parabiosis hosts, while MHC class II+CD11b+ cells in the L5 cord of wild type parabiosis mice had only a low number of CD45.1+ cells (*Figure 4B*). These results suggested that the majority of MHC class II+CD11b+ cells in the L5 cord of EAE remittent hosts originated from peripheral monocytes but not resident microglial cells.

Pain induction resulted in a strong accumulation of MHC class II+CD11b+ cells in the L5 cord, particularly at the ventral vessels (*Figure 4C*, ligation). We measured MFI and the number of MHC

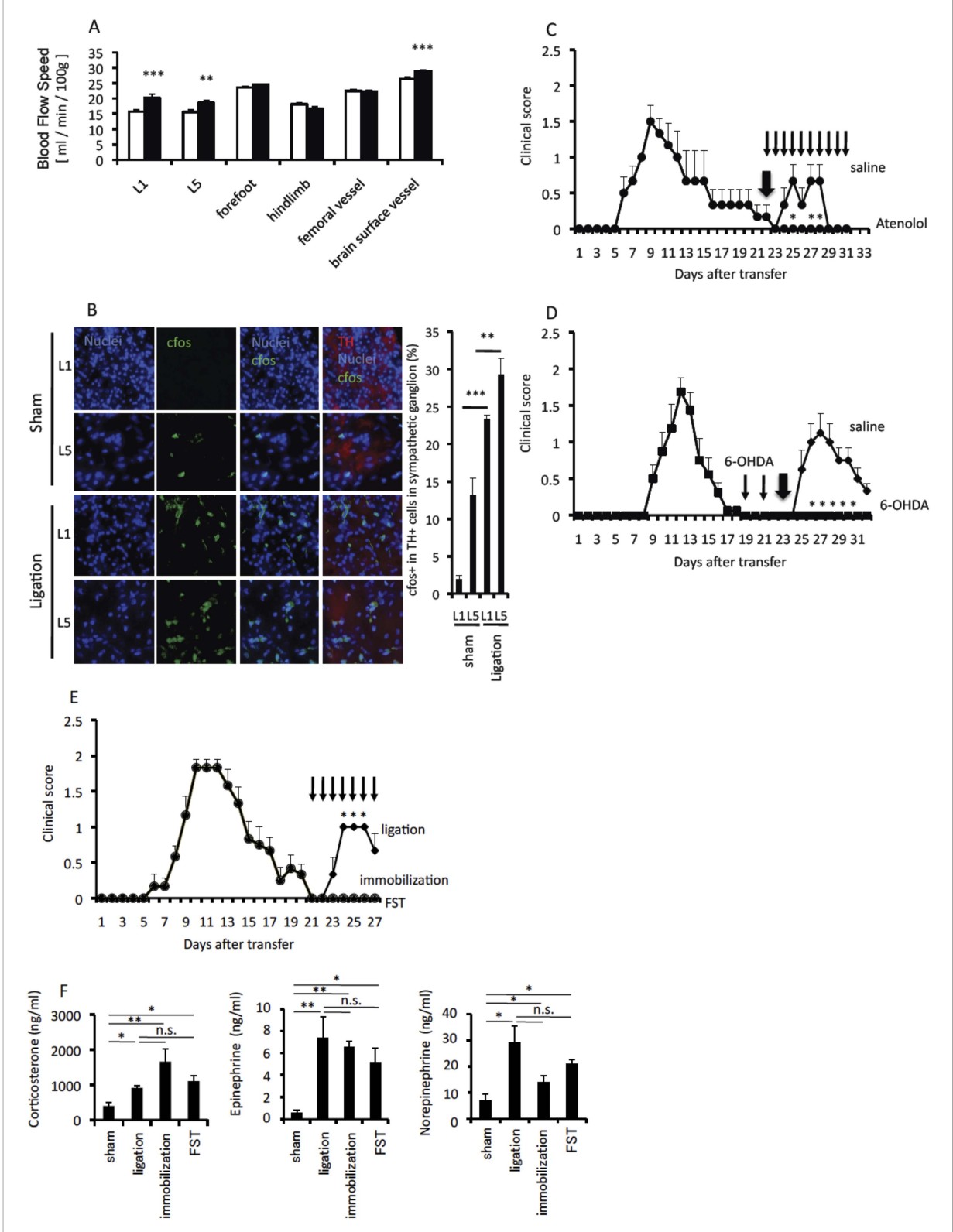

**Figure 3**. Sympathetic activation is involved in pain-mediated EAE relapse. Pain was induced in wild type C57BL/6 mice or EAE-recovered mice 20 days after pathogenic T cell transfer. (**A**) Blood flow speeds in blood vessels of various organs were measured 2 days after pain induction (closed bars) or in sham operated mice (open bars) (n = 3–5 per group). (**B**) Activation of sympathetic neurons in the first or fifth lumbar sympathetic ganglia was evaluated by cfos expression. Immunohistochemical staining for tyrosine hydroxylase (TH) is also shown (red) (n = 3 per group). Quantification of the histological

*Figure 3. continued on next page*

Figure 3. Continued

analysis of the somatosensory area based on cfos MFI (right). (C) Pathogenic CD4+ T cells isolated from EAE mice were intravenously transferred into wild type C57BL/6 mice. Pain was induced 22 days later (thick arrow), and EAE development was evaluated with or without atenolol treatment every day from day 22 in EAE-recovered mice (thin arrows) (n = 3–5 per group). (D) EAE development with or without 6-OHDA treatment at day 19 and day 21 (thin arrows) upon pain induction at day 23 (thick arrow) after T cell transfer in EAE-recovered mice (n = 3–5 per group). (E) EAE development with pain induction (diamonds), immobilization stress (circles), or forced swimming (FST) (triangles) every day from 21 days after T cell transfer in EAE-recovered mice (thin arrows) (n = 4–5 per group). (F) Serum concentrations of corticosterone, norepinephrine, and epinephrine with pain, immobilization stress or forced swimming 21 days after T cell transfer in EAE-recovered mice (n = 4–5 per group). Mean scores ± SEM are shown. *, p < 0.05, **, p < 0.01, ***, p < 0.001, n.s., not significant. Experiments were performed at least 3 times; representative data are shown.

The following figure supplement is available for figure 3:

Figure supplement 1. Mice treated with atenolol and 6-OHDA did not suppress cfos expression in neurons of the somatosensory area after pain induction.

class II+ cells in the dotted circles. Moreover, depletion of MHC class II+CD11b+ cells in the cisterna magna but not in the peritoneal cavity suppressed the accumulation of MHC class II+CD11b+ cells at L5 ventral vessels and the relapse development after pain induction (Figure 4C,D). These results demonstrated that the accumulation of MHC class II+CD11b+ cells, which were likely derived from the CNS side, at the L5 ventral vessels is important for EAE relapse.

We next investigated the connection between the sensory and sympathetic pathways, which is critical for the accumulation of MHC class II+CD11b+ cells at the L5 ventral vessels. Specific sensory afferents begin at the brainstem and travel via sympathetic efferent pathways through ACC. Neurons in the ACC are activated by NMDA receptors. We found an antagonist of NMDA receptors, MK801, injected to the ACC not only suppressed the expression of cfos but also the accumulation of MHC class II+CD11b+ cells at the ventral vessels in the L5 cord after pain induction (Figure 4E), while an agonist of NMDA receptors, L-Homocysteic acid, injected to the ACC enhanced the expression of cfos and the accumulation of MHC class II+CD11b+ cells at the ventral vessels in the L5 cord without ligation of the trigeminal nerves (Figure 4F). These results suggested that the pain-mediated neural activation was at least in part dependent on the ACC. Thus, it is possible that specific sensory afferents to the brainstem are important for sympathetic efferent pathways, which are critical for the accumulation of MHC class II+CD11b+ cells.

## Pain-mediated MHC class II+CD11b+ cell accumulation at L5 ventral vessels triggered EAE-relapse

We next investigated how pain stimulation accumulates MHC class II+CD11b+ cells at L5 ventral vessels. We hypothesized that sympathetic-mediated chemokine expressions might be involved. TH + sympathetic nerves were innervated and Creb molecules, a transcriptional factor of norepinephrine, were activated around L5 ventral vessels (Figure 5A,B). We also found chemokine expression around the ventral vessels (Figure 5C). Treatment with atenolol and 6-OHDA-mediated sympathectomy suppressed the accumulation of MHC class II+CD11b+ cells but not cfos expression in neurons of the somatosensory area after pain induction (Figure 5D,E, Figure 3—figure supplement 1).

To identify which specific chemokines are involved in the sympathetic-mediated accumulation of MHC class II+CD11b+ cells in the L5 cord, we first investigated the expression of chemokine receptors on MHC class II+CD11b+ cells in EAE-recovered mice, finding that the expression of CX3CR1 molecules and their ligand, CX3CL1, were increased particularly after pain induction (Figure 5F,G). Immunohistochemistry experiments confirmed that CX3CL1 molecules were found around the L5 ventral vessels (Figure 5C). Norepinephrine treatment increased CX3CL1 expression in primary CD11b+ cells (Figure 5H), while norepinephrine-mediated CX3CL1 expression was suppressed in CD11b+ cells derived from β1 and β2 receptor-deficient mice (Figure 5—figure supplement 1). Indeed, blockade of CX3CL1 by a neutralizing antibody in the CNS significantly suppressed the L5 accumulation of MHC class II+CD11b+ cells as well as the relapse development after pain induction (Figure 5I,J). These results demonstrated that pain-mediated CX3CL1 chemokine expression via the sympathetic pathway is critical for the accumulation of MHC class II+CD11b+ cells at the L5 ventral vessels of EAE-recovered mice.

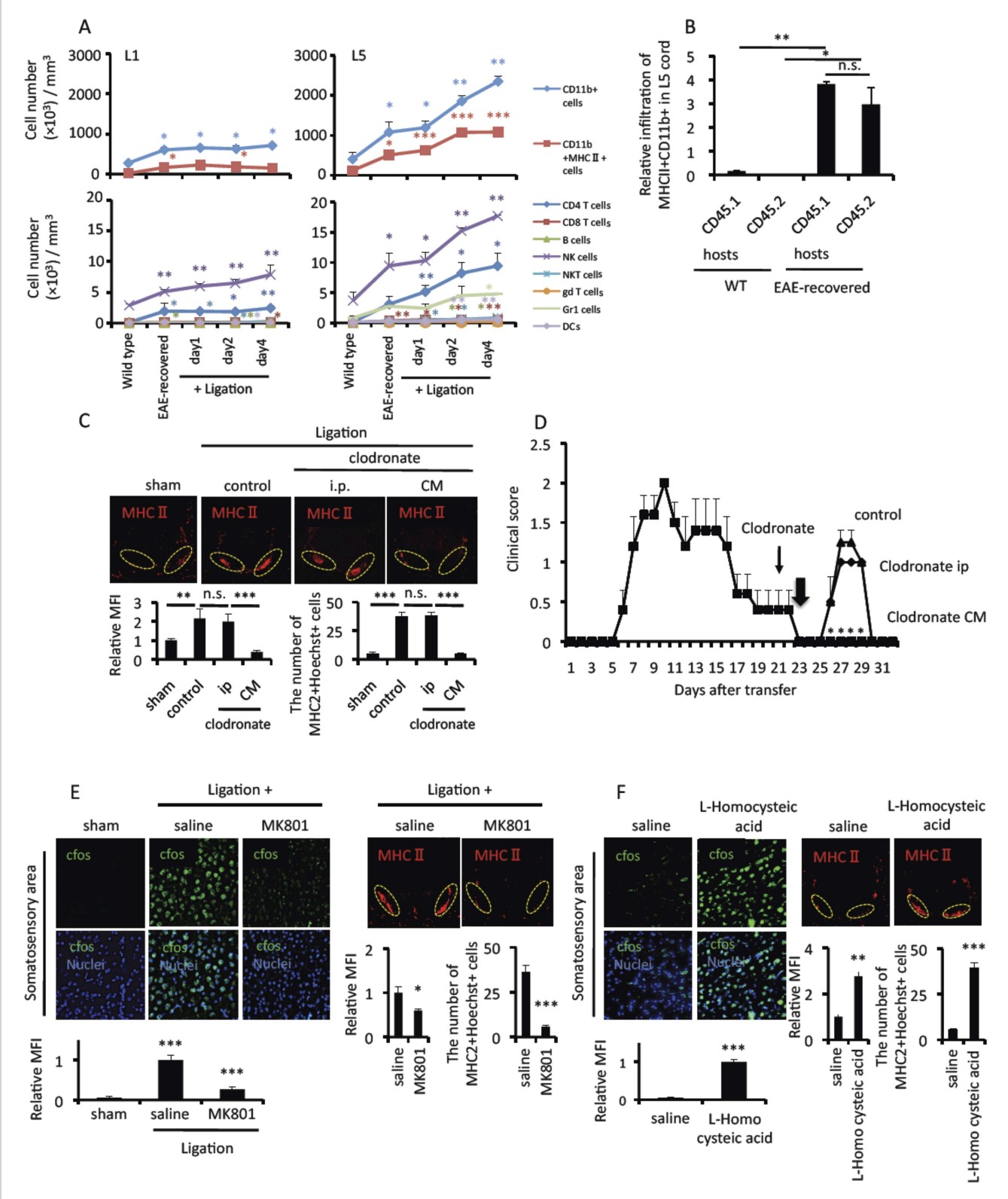

**Figure 4**. Pain induction accumulates MHC class II+CD11b+ cells at the ventral vessels of L5 via the anterior cingulate cortex in EAE-recovered mice. (**A**) Pathogenic CD4+ T cells isolated from EAE mice were intravenously transferred into wild type C57BL/6 mice. Pain was induced 20 days later in EAE-recovered mice (EAE-recovered mice). The L1 or L5 cord was isolated at days 0, 1, 2, and 4 after pain induction (n = 3–4 per group), and the corresponding number of cells was evaluated using a flow cytometer. Because L1 spinal cord has about sevenfold bigger volume than L5 cord according to an 11.7 tesla

*Figure 4. continued on next page*

**Figure 4. Continued**

MRI (*Mori et al., 2014*) (L1 average volume, 7.1 mm$^3$ and L5 average volume, 1.0 mm$^3$), immune cell numbers of L1 cord were divided by 7.1. Upper Figures, CD11b+ cells (blue); CD11b+MHC class II+ cells (red). Lower Figures, CD4+TCR+ cells (blue); CD8+TCR+ cells (red); B220+ cells (green); NK1.1+ TCR- cells (violet); NK1.1+TCR+ cells (sky blue); γδTCR+ cells (orange); Gr1+ cells (light green); CD11c+ (light violet). Cell numbers ± SEM are shown. Statistical comparisons were made with wild type. (**B**) Relative infiltration of MHC class II+CD11b+ cells in the L5 cord of C57BL/6-SJL mice based on parabiosis experiments using C57BL/6 mice (CD45.2+) and C57BL/6-SJL mice (CD45.1+) in the presence (EAE-recovered) or absence (WT) of EAE development induced by MOG-specific pathogenic CD4+ T cell transfer (n = 4–5 per group). 30 days after the T cell transfer, L5 cords of EAE-recovered C57BL/6-SJL hosts were evaluated. CD45.2+ cells are peripheral derived cells from another body. (**C**) Immunohistochemical staining for MHC class II in the L5 cord with or without pain induction and treatment with clodronate liposome administration in the cisterna magna (CM) or the peritoneal cavity (ip) (top) (n = 3 per group). Dotted circles show accumulated cells at the ventral vessels (top). Quantification of the histological analysis around the ventral vessels based on MFI in dotted circles (left bottom) and cell number of the accumulated MHC class II+ cells in dotted circles by using serial frozen sections stained with anti-MHC class II antibody and Hoechst (right bottom). (**D**) EAE development was evaluated with clodronate liposome administration into the cisterna magna (CM) or the peritoneal cavity (ip) day 21 after pathogenic T cell transfer (thin arrows) in the presence of pain induction in EAE-recovered mice (n = 3–5 per group). Pain was induced 23 days later (thick arrow). (**E**) Immunohistochemical staining for cfos in the somatosensory area of EAE-recovered mice with pain induction and MK801 administration to the somatosensory area (left top) (n = 3 per group). Quantification of the histological analysis of the somatosensory area based on cfos MFI (bottom). Immunohistochemical staining for MHC class II in the L5 cord with pain induction and MK801 administration to the somatosensory area (right top) (n = 3 per group). Dotted circles show accumulated cells at the ventral vessels (top). Quantification of the histological analysis around the ventral vessels based on MFI in dotted circles (bottom left) and cell number of the accumulated MHC class II+ cells in dotted circles by using serial frozen sections stained with anti-MHC class II antibody and Hoechst (bottom right). (**F**) Immunohistochemical staining for cfos in the somatosensory area of EAE-recovered mice with L-Homocysteic acid administration to the somatosensory area (left top) (n = 3 per group). Quantification of the histological analysis of the somatosensory area based on cfos MFI (left bottom). Immunohistochemical staining for MHC class II in the L5 cord with L-Homocysteic acid administration to the somatosensory area (right top) (n = 3 per group). Dotted circles show accumulated cells at the ventral vessels. Quantification of the histological analysis around the ventral vessels based on MFI in dotted circles (left bottom) and cell number of the accumulated MHC class II+ cells in dotted circles by using serial frozen sections stained with anti-MHC class II antibody and Hoechst (right bottom). Mean scores ± SEM are shown. *, p < 0.05; **, p < 0.01; ***, p < 0.001. Experiments were performed at least 3 times; representative data are shown.

MHC class II+CD11b+ cells in EAE-recovered mice expressed not only MHC class II molecules but also costimulatory molecules (*Figure 6A*) and had the ability to stimulate pathogenic CD4+ T cells without the addition of MOG peptide (*Figure 6B*), suggesting MHC class II+CD11b+ cells presented self-antigen peptides that stimulated pathogenic CD4+ T cells. We then investigated the relationship between MHC class II+CD11b+ cells and pathogenic CD4+ T cells in EAE relapse after pain induction. Anti-CD4 antibody treatment significantly suppressed the development of the EAE relapse (*Figure 6C*) despite the accumulation of MHC class II+CD11b+ cells (*Figure 6D*). Blockades of sympathetic activation and CX3CL1 expression suppressed the accumulation of not only MHC class II +CD11b+ cells, but also pathogenic CD4+ T cells at L5 ventral vessels (*Figure 6E*). These results are consistent with the idea that pain induction first causes an accumulation of MHC class II+CD11b+ cells at the ventral vessels of the L5 cord and then a MHC class II+CD11b+ cell-mediated accumulation and activation of pathogenic CD4+ T cells, the latter being also critical for the development of EAE relapse.

Because activated pathogenic CD4+ T cells express various cytokines including NFkB and STAT stimulators like IL-17 and IL-6, we considered whether chemokines were induced in the L5 ventral vessels via activation of the inflammation amplifier, a local chemokine inducer in endothelial cells. L5 ventral vessels indeed had increased expressions of various chemokines, including CCL20 and CCL5, in EAE-recovered mice particularly after pain induction (*Figure 6F*). NFkB and STAT3 molecules were simultaneously activated around L5 vessels after pain induction (*Figure 6F*). The blockade of IL-6, IL-17A, or CCL20 signaling suppressed the pain-mediated accumulation of pathogenic CD4+ T cells as well as the development of the EAE relapse without affecting MHC class II+CD11b+ cell accumulation (*Figure 6G–I* and data not shown). These results are consistent with the idea that excess expression of chemokines at L5 ventral vessels is critical for the development of EAE relapse via pathogenic CD4+ T cell accumulation.

## Discussion

We here demonstrated that pain-mediated sympathetic pathways induce relapse following sensory activation in EAE models. To induce pain, we mainly used nerve injury via the trigeminal nerves and painful agents including capsaicin injected into the whiskers or forefeet. To clarify if the relapse responses were due to neural signals and not pain-mediated systemic hormonal responses or stress-

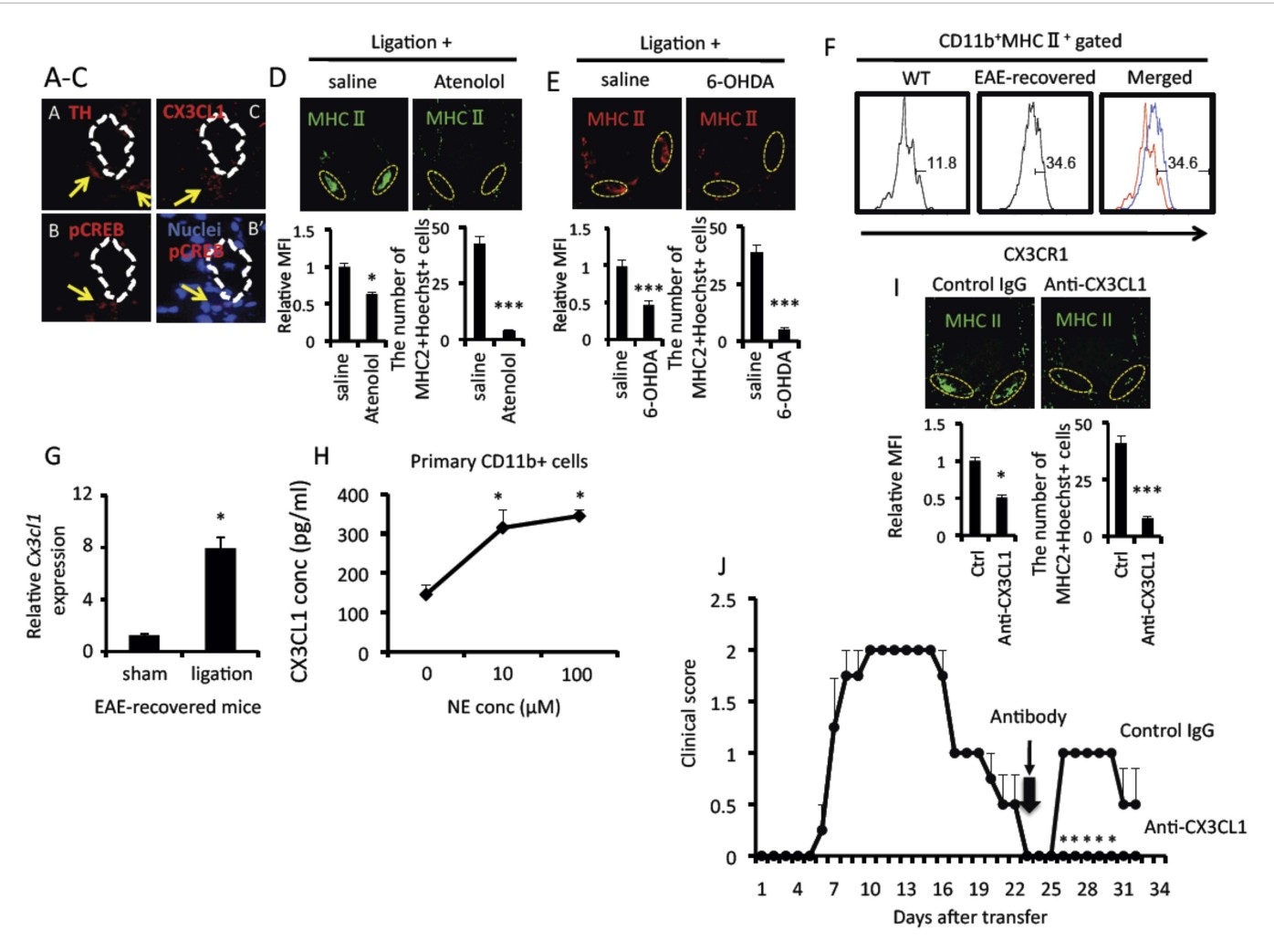

**Figure 5.** Pain mediated CX3CL1 expression via the sympathetic pathway is critical for the accumulation of activated MHC class II+CD11b+ cells at L5 ventral vessels in EAE-recovered mice. Immunohistochemical staining for (**A**) TH (tyrosine hydroxylase), phospho-CREB (**B**), merging of phospho-CREB with nuclei (**B'**) or CX3CL1 (**C**) in the ventral side of the L5 cord using serial sections (n = 3–5 per group). White dotted polygons indicate the shape of the L5 ventral vessel. Arrows show TH, phosphor-CREB, and CX3CL1 staining around the ventral vessels. (**D** and **E**) Immunohistochemical staining for MHC class II in the L5 cord with pain induction and treatment with atenolol (**D**) or 6-OHDA (**E**) (top) (n = 3 per group). Dotted circles show accumulated cells at the ventral vessels. Quantification of the histological analysis around the ventral vessels based on MFI in dotted circles (left bottom) and cell number of the accumulated MHC class II+ cells in dotted circles by using serial frozen sections stained with anti-MHC class II antibody and Hoechst (right bottom). (**F**) The expression of CX3CR1 on CD11b+MHC class II+ cells of the spinal cord in healthy wild type mice (WT, left and red in right) and EAE-recovered mice (EAE-recovered, middle and blue in right) (n = 4 per group). (**G**) CX3CL1 mRNA expression in the ventral vessels was investigated by real time PCR with or without pain induction in EAE-recovered mice (n = 3–5 per group). (**H**) CX3CL1 expression was enhanced in the presence of norepinephrine. Primary CD11b+ cells in wild type mice (n = 2) were stimulated with norepinephrine for 24 hr. Culture supernatants were collected and assessed using an ELISA specific for mouse CX3CL1. (**I**) Immunohistochemical staining for MHC class II in the L5 cord with pain induction and treatment with an anti-CX3CL1 antibody, which was injected into the cisterna magna (top) (n = 3–5 per group). Dotted circles show accumulated cells at the ventral vessels. Quantification of the histological analysis around the ventral vessels based on MFI in dotted circles (left bottom) and cell number of the accumulated MHC class II+ cells in dotted circles by using serial frozen sections stained with anti-MHC class II antibody and Hoechst (right bottom). (**J**) Pathogenic CD4+ T cells isolated from EAE mice were intravenously transferred into wild type C57BL/6 mice. Pain was induced 23 days later (thick arrow), and EAE development was evaluated with or without anti-CX3CL1 antibody injection into the cisterna magna in EAE-recovered mice on day 23 (thin arrow) (n = 4–5 per group). Mean scores ± SEM are shown. *, p < 0.05; ***, p < 0.001. Experiments were performed at least 3 times; representative data are shown.

The following figure supplement is available for figure 5:

**Figure supplement 1.** CX3CL1 expression in CD11b+ cells was enhanced in the presence of norepinephrine in a manner dependent on β1 and β2 receptors.

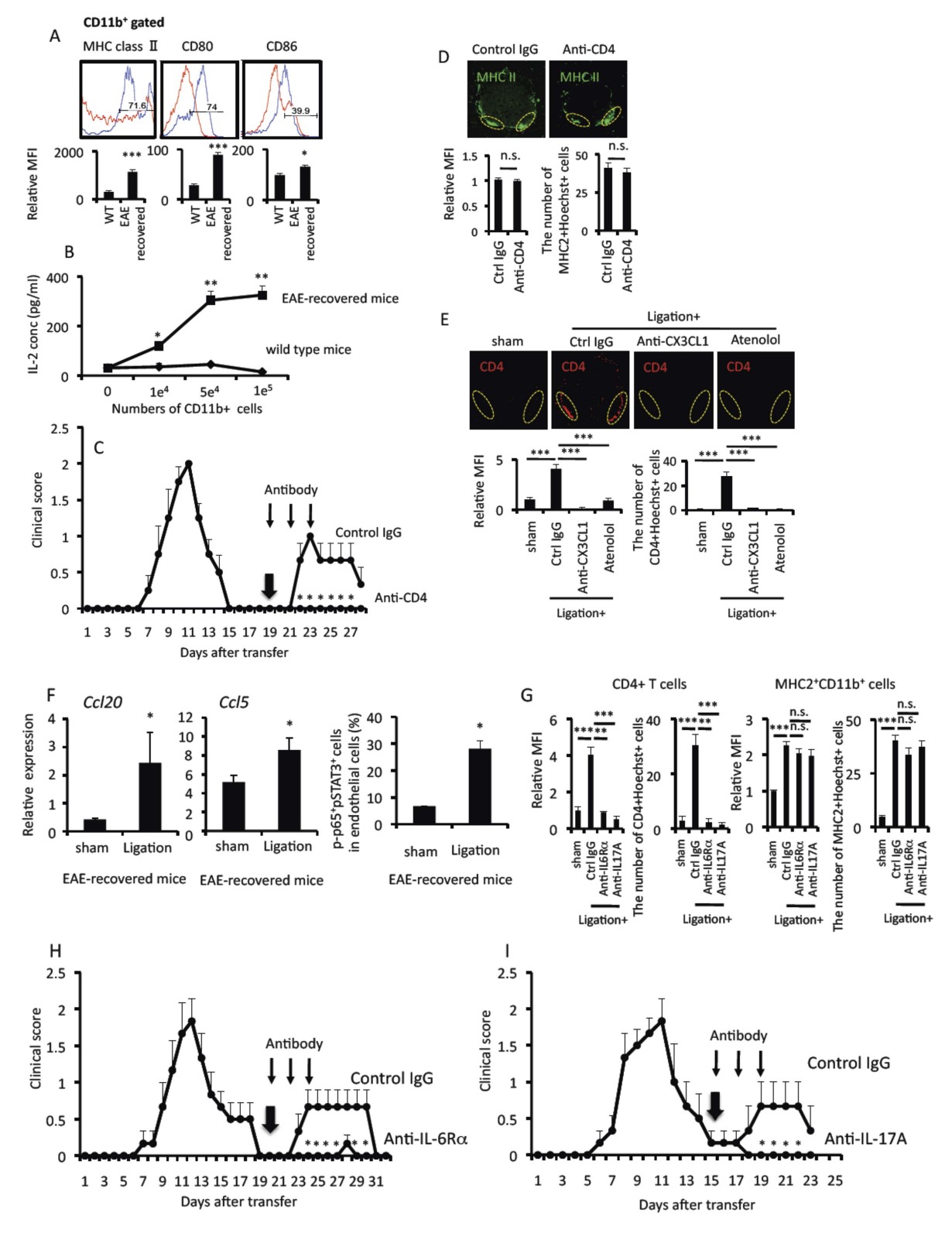

**Figure 6.** Pain-mediated accumulation of MHC class II+CD11b+ cells and CD4+ T cells at L5 ventral vessels is important for the development of EAE relapse. (**A**) The expression of MHC class II, CD80, and CD86 on CD11b+ cells in the spinal cord of wild type mice (red) and EAE-recovered mice (blue) (top) (n = 4 per group). Mean Fluorescence Intensity (bottom). (**B**) CD11b+ cells isolated from EAE-recovered mice but not wild type mice have the potential of antigen presentation to CD4+ T cells without peptide addition (n = 2 per group). Culture supernatants were collected and assessed using an

*Figure 6. Continued*

ELISA specific for mouse IL-2. (**C**) Pathogenic CD4+ T cells isolated from EAE mice were intravenously transferred into wild type C57BL/6 mice. EAE development was evaluated 19 days later with or without intra-peritoneal injection of an anti-CD4 antibody (days 19, 21, and 23, thin arrows) and pain induction (day 19, thick arrow) in EAE-recovered mice (n = 4–5 per group). (**D**) Immunohistochemical staining for MHC class II in the L5 cord with pain induction (8 hr) and treatment with anti-CD4 antibody (using the same hosts as (**C**)) (top) (n = 4–5 per group). Dotted circles show accumulated cells at the ventral vessels. Quantification of the histological analysis around the ventral vessels based on MFI in dotted circles (bottom left) and cell number of the accumulated MHC class II+ cells in dotted circles by using serial frozen sections stained with anti-MHC class II antibody and Hoechst (bottom right). (**E**) Immunohistochemical staining for CD4 in the L5 cord with or without pain induction (8 hr) and treatment with anti-CX3CL1 antibody or atenolol (top) (n = 3–5 per group). Dotted circles show accumulated cells at the ventral vessels. Quantification of the histological analysis around the ventral vessels based on MFI in dotted circles (left bottom) and cell number of the accumulated CD4+ cells in dotted circles by using serial frozen sections stained with anti-CD4 antibody and Hoechst (right bottom). (**F**) CCL20 and CCL5 mRNA expressions at L5 ventral vessels was evaluated with or without pain induction in EAE-recovered mice (left and middle) (n = 3 per group). Quantification of the histological analysis for p65+pSTAT3+ endothelial cells around ventral vessels based on serial frozen sections in EAE-recovered mice with or without pain induction (right) (n = 3 per group). (**G**) Immunohistochemical staining for CD4 and MHC class II in the L5 cord with pain induction and treatment with anti-IL-6Rα antibody or anti-IL-17A antibody (n = 4–5 per group). Quantifications of the histological analysis around ventral vessels based on serial frozen sections (CD4 [left] and MHC class II [right]) are shown. (**H**) Pathogenic CD4+ T cells isolated from EAE mice were intravenously transferred into wild type C57BL/6 mice. EAE development was evaluated from day 0–32 after pathogenic T cell transfer with or without anti-IL-6 receptor α antibody treatment (days 20, 22, and 24, thin arrows) upon pain induction (day 20, thick arrow) in EAE-recovered mice (n = 4–5 per group). (**I**) Pathogenic CD4+ T cells isolated from EAE mice were intravenously transferred into wild type C57BL/6 mice. EAE development was evaluated 15 days later with or without intra-peritoneal injection of an anti-IL-17A antibody treatment (days 15, 17, and 19, thin arrows) or pain induction (day 15, thick arrow) in EAE-recovered mice (n = 4–5 per group). Mean scores ± SEM are shown. *, p < 0.05; **, p < 0.01; ***, p < 0.001. Experiments were performed at least 3 times; representative data are shown.

The following figure supplement is available for figure 6:

**Figure supplement 1**. MHC class II+CD11b+ cells in the CNS of EAE-recovered mice expressed TNFα and IL-1β.

mediated events, we first investigated sensory and sympathetic pathways, finding that sensory neurons expressing TRPV1 and/or Nav1.8, sensory-sympathetic connections in the ACC, and sympathetic neurons were involved in the response. On the other hand, EAE mice under stress conditions did not develop EAE relapse, although the stress inductions did increase serum corticosterone, norepinephrine, and epinephrine similarly to pain induction (*Figure 3E,F*). These data suggest specific sensory-sympathetic signals triggered by pain induction plays a role in the development of EAE relapse.

We could divide the relapse responses into at least four steps. The first step is pain-mediated sensory activation followed by sympathetic activation. We found various sensory neurons are activated after pain induction (*Figure 2B*). TRPV1 deficiency or treatment with a Nav1.8 blocker, A803467, suppressed the development of the EAE relapse and its related events including the accumulation of MHC class II+CD11b+ cells around L5 ventral vessels and cfos expression at the neurons of the somatosensory area after pain induction (*Figure 2C,D*, and *Figure 2—figure supplement 1*). Consistent with these results, we showed that the injections of a TRPV1 agonist, capsaicin in EAE-recovered mice induced the development of EAE relapse and related events, phenotypes that are consistent with those induced by pain in the trigeminal nerves (*Figure 1F* and *Figure 1—figure supplement 2*).

The second step is pain-mediated sympathetic activation. This activation triggers the regional expression of the chemokine CX3CL1 followed by the accumulation of MHC class II+CD11b+ cells at the L5 ventral vessels. Because Creb was activated around the L5 ventral vessels and atenolol and 6-OHDA-mediated sympathectomy suppressed the accumulation of MHC class II+CD11b+ cells (*Figure 5B,D,E*), we further concluded that the norepinephrine pathway plays a role. CD11b+ cells in the CNS are likely to express CX3CL1, because these cells induced CX3CL1 after norepinephrine stimulation, at least in vitro (*Figure 5H* and *Figure 5—figure supplement 1*). The number of MHC class II+CD11b+ cells was higher in the L5 cord than other cords, particularly in EAE-recovered mice, even without pain induction (*Figure 4A*). The L5 specificity of the relapse development may be because the L5 cord is the site of initial inflammation and is the tissue most damaged during the primary development of EAE. Moreover, it is known that a certain level of the sympathetic pathway is activated by anti-gravity responses even at steady state (*Arima et al., 2012*). Therefore, we hypothesized that a low but

sufficient level of sympathetic activation may maintain a low level of the inflammation state in the L5 cord to accumulate MHC class II+CD11b+ cells in the L5 cord of EAE-recovered mice and develop EAE relapse after pain induction.

The third step for EAE relapse is the accumulation of pathogenic CD4+ T cells at the L5 ventral vessels. This step is most likely dependent on self-antigen presentation via MHC class II+CD11b+ cells, because these cells stimulated MOG-specific pathogenic CD4+ T cells (*Figure 6B*).

The fourth and final step is mediated by excessive chemokine expressions and is most likely triggered by the activation of the accumulated pathogenic CD4+ T cells in the third step. Pathogenic CD4+ T cells transferred into mice include Th17 and Th1 cells and express various cytokines beyond IL-17 and IFNγ, including IL-6, TNFα, etc., all of which are stimulators of NFkB and STATs (*Ogura et al., 2008*; *Lee et al., 2012*; *Murakami et al., 2013*; *Atsumi et al., 2014*). Moreover, we have previously shown that endothelial cells express IL-6 after norepinephrine stimulation (*Arima et al., 2012*) and MHC class II+CD11b+ cells in the CNS of EAE-recovered mice expressed various cytokines (*Figure 6—figure supplement 1*). Therefore, we suggest that the local chemokine expression at L5 ventral vessels via various cytokines subsequently enhances an accumulation of immune cells, as shown in *Figure 4A*. Indeed, we found blockades of IL-6, IL-17A, or CCL20 signals suppressed EAE relapse (*Figure 6H,I*, and data not shown). The excessive chemokine induction results in excess immune cell migration, which compromises local homeostasis and can trigger inflammatory diseases. Thus, accumulation of both MHC class II+CD11b+ cells and pathogenic CD4+ T cells at the ventral vessels of the L5 cord stimulate the excessive expression of chemokines in non-immune cells to trigger the EAE relapse.

Although the flow cytometry data in *Figure 4A* showed that all cell types were recruited at similar rates and did not accumulate in distinct waves, it should be pointed out that the sensitivity of the FACS analysis is low with regards to the MHC class II+CD11b+ cells and CD4+ T cells found in the immunohistochemistry. Therefore, FACS analysis will not reveal sequential steps. The results of the blocking experiments, however, do show obvious four steps: Regarding the first step, blockades of sensory pathways by using A803467, TRPV1-deficient hosts, and MK801 at the ACC region suppressed somatosensory activations, sympathetic activations, and MHC class II+CD11b+ cell accumulation in the ventral vessels, and EAE-relapse. Regarding the second step, we used atenolol and 6-OHDA to suppress sympathetic activation and MHC class II+CD11b+ cell accumulation in the ventral vessels and/or EAE-relapse. However, this treatment did not affect sensory-mediated somatosensory activations, suggesting that sympathetic activation is a downstream event after sensory activation. Also regarding the second step, blockade of CX3CL1 did suppress MHC class II+CD11b+ cell accumulation in the ventral vessels and EAE-relapse but not somatosensory activations and sympathetic activations, suggesting that CX3CL1 expression is a downstream event after sensory and sympathetic activation. Regarding the third step, anti-CD4 antibody application suppressed CD4+ T cell accumulation in the ventral vessels and EAE-relapse but not somatosensory activations, sympathetic activations, or MHC class II+CD11b+ cell accumulation in the ventral vessels, suggesting that CD4+ T cell accumulation is a downstream event of sensory and sympathetic activation and of MHC class II+CD11b+ cell accumulation in the ventral vessels. MHC class II+CD11b+ cell accumulation in the ventral vessels as observed within 8 hr. Regarding the fourth step, blockades of IL-17 and IL-6 signal by neutralizing antibodies suppressed EAE-relapse but not somatosensory activations, sympathetic activations, MHC class II+CD11b+ cell accumulation in the ventral vessels, or CD4+ T cell accumulation, suggesting that the inflammation amplifier activation by cytokines including IL-17 and IL-6 is a downstream event after sensory and sympathetic activation, MHC class II+CD11b+ cell accumulation in the ventral vessels, and pathogenic Th17 cell accumulation in the ventral vessels. These results clearly suggested there are the four-steps that occur sequentially and are critical for the development of pain-mediated EAE relapse in our EAE models.

Regarding the induction of different chemokine-mediated axes by different sympathetic neurons (pain in this paper vs anti-gravity in the previous paper [*Arima et al., 2012*]), we should point out that we used different hosts in these two cases. We used normal mice for anti-gravity experiments, but EAE-recovered mice for pain-mediated ones. In other words, the numbers of MHC class II+CD11b+ cells are completely different: normal mice have almost no MHC class II+CD11b+ cells in the CNS, while EAE-recovered mice have many MHC class II+CD11b+ cells in the CNS particularly in the L5 cord. In EAE-recovered mice, a regional norepinephrine output via pain inductions functions on MHC class II+CD11b+ cells as well as endothelial cells around the ventral vessels of the spinal cords, causing

the expression of CX3CL1 and accumulation of MHC class II+CD11b+CX3CR1+ cells in a manner dependent on the CX3CL1-CX3CR1 axis. These accumulated MHC class II+CD11b+CX3CR1+ cells present autoantigens to pathogenic Th17 cells located in the blood stream. On the other hand, in normal mice, a regional norepinephrine output via anti-gravity responses functions just on endothelial cells because of there are almost no MHC class II+CD11b+ cells in the CNS. This case results in CCL20 expression, which causes the accumulation pathogenic Th17 cells that express the CCR6 receptor, IL-17 and IL-6.

We hypothesized pain-mediated sympathetic signaling acts via a high concentration of norepinephrine at the sympathetic neurovascular connection, but not through the systemic induction of hormones such as epinephrine and norepinephrine, which play a major role in the development of EAE relapse, because neither an accumulation of MHC class II+CD11b+ cells at L5 ventral vessels nor EAE relapse in remittent mice that had immobilization stress or forced swimming were observed, even though serum norepinephrine and epinephrine increased just like in the pain-induction condition, which did induce MHC class II+CD11b+ cell accumulation as well as EAE relapse. However, epinephrine and norepinephrine induced CX3CL1 expression dose-dependently in MHC class II+CD11b+ cells, which are also present in spleen and lymphonodes, and endothelial cells and fibroblasts induced various chemokines, including CCL20, after stimulation by epinephrine and norepinephrine, particularly in the presence of NFkB and STAT3 activation (*Arima et al., 2012*). Therefore, we hypothesized that systemic epinephrine/norepinephrine molecules induced a systemic pro-inflammatory state via chemokine expression.

MS GWAS analysis in 2011 showed a clear link between MS patients and the MHC class II-CD4+ T cell axis, but also suppressive effects on the MHC class I-CD8+ T cell axis (*Consortium, 2011*), suggesting that MS is a CD4 genetic disease but not CD8 one. On the other hand, it is known that MS active lesions contain many CD8+ T cells sometimes having features of local antigen reactivity (*Tsuchida et al., 1994*; *Dressel et al., 1997*; *Crawford et al., 2004*). We therefore hypothesized that CD8+ T cell accumulation is mediated by inflammation responses primarily induced by autoreactive CD4+ T cells. Consistently, our results from transfer EAE also showed the accumulations of various immune cells including CD8+ T cells in L5 cord.

It is well known that many patients with MS experience pain (*Thompson et al., 2010*). Kalia and O'Connor found that the proportion of patients with a progressive course of MS trend higher among patients with chronic types of pain (*Kalia and O'Connor, 2005*). It was also reported that central pain is the first and sole symptom of MS relapse in some patients (1.6% of all patients, 6% of patients with central pain) (*Osterberg et al., 2005*). At the same time, several reports have failed to show a clear relationship between pain and disease duration and/or course (*Beiske et al., 2004*; *Kalia and O'Connor, 2005*; *Michalski et al., 2011*). Thus, there does not seem to be a direct correlation clinically between pain intensity and MS disability or lesional load. We hypothesize that there are several reasons why we did not observe a clear relationship between pain and MS progression, which we explain in view of the above four steps. In the first step, differences in pain sensitivity and sensory activation among individuals complicate the investigation. It is known that loss of pain sensitivity may vary with the region or size of the affected site between patients, such that a bigger affected site correlates with less sensitivity in MS patients (*Huber et al., 1988*). In addition, depression occurs with high frequency in MS patients (*Chwastiak et al., 2002*; *Siegert and Abernethy, 2005*), and this might also affect sensation. In the second step, the degree of sympathetic activation should vary among individual patients following the first step, as too might the connection between the sensory and sympathetic pathways. Moreover, depression in MS patients also affects the sympathetic pathway (*Guinjoan et al., 1995*; *Rumsfeld and Ho, 2005*). In the third and fourth steps, which are mediated by autoreactive CD4+ T cell-mediated inflammation amplifier activation following MHC class II+CD11b+ cell accumulation, different affected regions in quantity and/or quality between MS patients due to differences in the autoantigen distribution recognized by autoreactive CD4+ T cells and different neural activations in each patient should disrupt the relationship between pain and MS progression. Neural activations, which are critical for the establishment of autoreactive CD4+ T cell gateways in the vessels, might also be affected by various factors in each patient. These reasons might explain the disrupted relationship between pain sensation and MS progression. In our experiments, however, we managed to control the majority of these variations in MS mouse models. Thus, based on our analysis, we concluded that pain-mediated neural signal induces EAE relapse.

We propose at least three factors as important for determining the sites of relapse in MS patients: (1) the existence of excess MHC class II+CD11b+ cells at a particular site, (2) activation of a sympathetic pathway that accumulates MHC class II+CD11b+ cells in certain blood vessels, and (3) the presence of organ-specific autoreactive CD4+ T cells in the periphery. Even when MHC class II+CD11b+ cells are accumulated in a specific vessels, the inflammation is not induced without autoreactive CD4+ T cells, which recognized some autoantigens on the accumulated MHC class II+CD11b+ cells. We assume the sites of excess MHC class II+CD11b+ cells and sympathetic activations vary among patients as too the antigens recognized by autoreactive T cells, which would explain the variation in MS relapse regions. In other words, if we correctly detect the site of MHC class II+CD11b+ cell accumulation, the sites of sympathetic activation, and the presence of autoreactive CD4+ T cells in the periphery, we might be able to discern where the MS relapse begins and how to regulate it.

## Materials and methods

### Mouse strains

C57BL/6 mice were purchased from Japan SLC (Tokyo, Japan), Adrb1-deficient mice, Adrb2-deficient mice and TRPV1-deficient mice were purchased from The Jackson Laboratory (Bar Harbor, ME), C57BL/6-SJL mice were purchased from Taconic (Germantown, NY), and SJL/J mice were purchased from Charles River (Yokohama, Japan). All mice were maintained under specific pathogen-free conditions according to the protocols of the Hokkaido University and the Osaka University Medical Schools.

### Passive transfer of pathogenic CD4+ T cells from mice to induce EAE

EAE induction was performed as described previously (*Ogura et al., 2008*; *Arima et al., 2012*). Briefly, C57BL/6 mice were injected with a MOG (35–55) peptide (Sigma–Aldrich, Tokyo) in complete Freund's adjuvant (Sigma–Aldrich) at the base of the tail on day 0 followed by intravenous injection of pertussis toxin (Sigma–Aldrich) on days 0, 2, and 7. On day 9, CD4$^+$ T cells from the resulting mice were sorted using anti-CD4 microbeads (Miltenyi Biotec, Tokyo). The resulting CD4$^+$ T cell-enriched population ($4 \times 10^6$ cells) was cocultured with rIL-23 (10 ng/ml; R&D Systems, Minneapolis, MN) in the presence of MOG peptide-pulsed irradiated splenocytes ($1 \times 10^7$ cells) for 2 days. Cells ($1.5 \times 10^7$ cells) were then injected intravenously into wild type mice. Clinical scores were measured as described previously (*Ogura et al., 2008*; *Arima et al., 2012*).

### Relapsing-remitting EAE induction

For the induction of relapsing-remitting EAE, SJL mice were injected with a proteolipid protein (PLP) (139–151) peptide (TOCRIS, Tokyo) in complete Freund's adjuvant (Sigma–Aldrich) at the base of the tail on day 0 followed by intravenous injection of pertussis toxin (Sigma–Aldrich) on days 0, 2, and 7. The severity of EAE was evaluated in a blinded fashion using following scale as previously described (*Moriya et al., 2008*). 0, normal; 1, limp tail; 2, mild paraparesis of the hind limbs with unsteady gait; 3, moderate paraparesis with preservation of voluntary movement; 4, paraplegia; 5, moribund.

### Pain induction

Animal treatments were performed according to the Guidelines of the International Association for the Study of Pain (*Zimmermann, 1983*). Before the experiments, the animals were allowed to habituate to the housing facility for 3 days. Ligation of the middle branch of trigeminal neurons and the von Frey test were performed as described previously (*Figure 1—figure supplement 1*) (*Krzyzanowska et al., 2011*). In brief, surgical procedures were performed under anesthesia. The right trigeminal neuron was exposed and loosely ligated around the distal part of the nerve with a polyglycolic acid suture (Akiyama MEDICAL CO., Ltd, 6-0, Sapporo, Japan), as previously described in rats (*Vos et al., 1994*). Care was taken not to cause excessive compression of the nerve, as this could hamper the emergence of allodynia (*Martin and Avendano, 2009*). In sham surgeries (*n* = 5), the trigeminal neuron was exposed but left untouched. Tactile allodynia was measured using von Frey filaments.

## Cell isolation from spinal cords

After perfusion with ice cold PBS, spinal cords were dissected and enzymatically digested using the Neural Tissue Dissociation Kit (P) (Miltenyi Biotec, Tokyo). CD11b+ cells were isolated by suspending them in MACS buffer and staining them with anti-CD11b microbeads (Miltenyi Biotec) followed by separation in a magnetic field using an MS column (Miltenyi Biotec).

## Histological analysis

Spines were harvested and embedded in SCEM compound (SECTION-LAB Co. Ltd., Hiroshima, Japan) and prepared as sections using the microtome device CM3050 (Leica Microsystems, Tokyo) and macrotome device CM3600XP (Leica Microsystems) with Cryofilm type IIC9 (SECTION-LAB Co. Ltd.). The resulting sections were stained with hematoxylin/eosin or immunohistochemical staining and analyzed with a BZ-9000 microscope (KEYENCE, Osaka, Japan). Analysis was performed by HS ALL software in one fluorescence microscope BZ-II analyzer (KEYENCE). Frozen sections (10 μm) were prepared according to a published method (*Kawamoto, 2003*; *Arima et al., 2012*).

## Antibodies and reagents

The following antibodies were used for the flow cytometry analysis: FITC-conjugated anti-CD19 (eBioscience, Tokyo), anti-Gr1 (eBioscience), anti-CD80 (eBioscience), anti-CD45.2 (eBioscience), PE-conjugated anti-TCRβ (eBioscience), anti-NK1.1 (eBioscience), anti-I-A/I-E (BioLegend, Tokyo), anti-CD86 (eBioscience), anti-CD193 (CCR3) (BioLegend), anti-CMKLR1 (eBioscience), PE-Cy7-conjugated anti-CD8 (eBioscience), anti-CD3 (eBioscience), anti-CD45.1 (eBioscience), eFluor450-conjugated anti-CD45 (eBioscience), anti-CD4 (eBioscience), APC-conjugated anti-CD4 (BioLegend), anti-γδTCR (eBioscience), anti-CD11c (eBioscience), anti-I-A/I-E (BioLegend), anti-CD45.2 (eBioscience), biotin-conjugated anti-CD11b (eBioscience), anti-CX3CR1 (Abcam, Tokyo), anti-CD195 (CCR5) (eBioscience), anti-CD197 (CCR7) (eBioscience), anti-CD183 (CXCR3) (eBioscience), anti-CD184 (CXCR4) (eBioscience), and anti-CD185 (CXCR5) (eBioscience). The following antibodies were used for immunohistochemistry: anti-phospho-STAT3 (Tyr705, D3A7), anti-phospho-NFkB anti-phospho-p65, anti-phospho-CREB (Cell Signaling, Tokyo), anti-tyrosine hydroxylase (Abcam), anti-cFos (Sigma–Aldrich), control rabbit IgG (DA1E) (Cell Signaling), anti-CX3CL1 (Abcam), anti-Nav1.8 antibody (Abcam), anti-VR1 antibody (Abcam), anti-NeuN antibody (Millipore, Tokyo), biotin-conjugated anti-CD4 (BioLegend), anti-CD11b (eBioscience), anti-I-A/I-E (BioLegend), anti-CD86 (BioLegend), Alexa Fluor 488 goat anti-rabbit IgG (H + L), Alexa Fluor 546 goat anti-rabbit IgG (H + L), Alexa Fluor 647 goat anti-chicken IgG (Invitrogen, Tokyo), and Streptavidin Alexa Fluor 546 conjugate (Invitrogen). The following antibodies were used for in vivo neutralization: purified anti-mouse CCL20 mAb, anti-mouse IL-17 Ab, and anti-CX3CL1 Ab (R&D Systems). The anti-CD4 antibody was purified as described previously (*Ueda et al., 2006*). The anti-IL-6 receptor antibody was obtained from Chugai Pharmaceutical Co (Tokyo, Japan). Atenolol, capsaicin, 6-Hydroxydopamin hydrochloride, A-803467, Norepinephrine, MK801, and L-Homocysteic acid were purchased from Sigma–Aldrich. Gapapentin was purchased from Tokyo Chemical Industry (Tokyo). Pregabalin was purchased from Taconic (Tokyo). The VECTASTAIN Elite ABC Rabbit IgG Kit and the DAB Peroxidase Substrate Kit were purchased from Vector Laboratories (Burlingame, CA).

## ELISA and EIA

CX3CL1 and IL-2 levels in cell culture supernatants were determined using ELISA kits from R&D Systems and eBiosciences, respectively. Norepinephrine and epinephrine levels in serum were determined using EIA kits from Labor Diagnostika Nord (Nordhorn, Germany) and corticosterone levels in serum using EIA kits from Abnova (Taipei, Taiwan).

## Flow cytometry

To generate single cell suspension, spinal cords were dissected and enzymatically digested using the Neural Tissue Dissection Kit (Miltenyi Biotec), and $10^6$ cells were incubated with fluorescence-conjugated antibodies for 30 min on ice for cell surface labeling. The cells were then analyzed with cyan flow cytometers (Beckman Coulter, Tokyo). The collected data were analyzed using Summit software (Beckman Coulter) and/or Flowjo software (Tree Star, Ashland, OR).

## Immunohistochemistry

Immunohistochemistry was performed as described previously with slight modifications (*Lee et al., 2012*).

## Laser micro-dissection

Approximately 100 frozen sections (15 μm) were fixed with acetic acid/ethyl alcohol (1:19) for 15 min followed by PBS-washing for 10 min. Tissues around the ventral vessels in the sections were collected by a laser micro-dissection device, DM6000B (Leica Microsystems), and prepared for total RNA measurements by the GenElute Mammalian Total RNA Kit (Sigma–Aldrich) and Ethachinmate (Nippon Gene, Tokyo).

## Real-time PCRs

A GeneAmp 5700 sequence detection system (ABI, Tokyo), KAPA PROBE FAST ABI Prism qPCR Kit (Kapa Biosystems, Boston, MA), and KAPA SYBR FAST ABI Prism qPCR Kit (Kapa Biosystems) were used to quantify the levels of CCL20 mRNA, CCL5 mRNA, CX3CL1 mRNA, IL-1β mRNA, TNFα mRNA, and HPRT mRNA. The PCR primer pairs used for real-time PCRs using KAPA PROBE FAST ABI Prism qPCR Kit were as follows: mouse HPRT primers, 5′-AGCCCCAAAATGGTTAAGGTTG-3′ and 5′-CAAGGGCATATCCAACAACAAAC-3′, probe, 5′-ATCCAACAAAGTCTGGCCTGTATCCAACAC-3′; mouse CCL20 primers, 5′-ACGAAGAAAAGAAAATCTGTGTGC-3′ and 5′-TCTTCTTGACTCTTAGGCTGAGG-3′, probe, AGCCCTTTTCACCCAGTTCTGCTTTGGA; mouse CX3CL1 primers, 5′-CGTTCTTCCATTTGTGTACTCTGC-3′ and 5′-AGCTGATAGCGGATGAGCAAAG-3′, probe, 5′-TCAGCACCTCGGCATGACGAAATGCG-3′; and mouse CCL5 primers, 5′-CTCCCTGCTGCTTTGCCTAC-3′ and 5′-CGGTTCCTTCGAGTGACAAACA-3′, probe, 5′-TGCCTCGTGCCCACGTCAAGGAGTATT-3′. The PCR primer pairs used for real-time PCRs using the KAPA SYBR FAST ABI Prism qPCR Kit were as follows: mouse HPRT primers, 5′-GATTAGCGATGATGAACCAGGTT-3′ and 5′-CCTCCCATCTCCTTCATGACA-3′; mouse IL-1β primers, 5′-TTGACGGACCCCAAAAGATG-3′ and 5′-TGGACAGCCCAGGTCAAAG-3′; and mouse TNFα primers, 5′-TACTGAACTTCGGGGTGATCGGTCC-3′ and 5′-CAGCCTTGTCCCTTGAAGAGAACC-3′. The conditions for real-time PCR were 40 cycles at 95°C for 3 s followed by 40 cycles at 60°C for 30 s. The relative mRNA expression levels were normalized to the levels of HPRT mRNA.

## Treatments of antibodies and reagents

In some experiments, one of anti-CCL20 antibody (200 μg/mouse), anti-IL-17A antibody (100 μg/mouse), anti-IL-6 receptor-α antibody (500 μg/mouse), anti-CD4 antibody (200 μg/mouse), or atenolol (500 μg/mouse) along with clodronate liposomes 300 (1 mg/mouse), Gabapentin (1 mg/mouse), and A-803467 (200 μg/mouse) were intraperitoneally injected into EAE-recovered mice that had pain induced at the same time. Gabapentin (1 mg/mouse) and Pregabalin (250 μg/mouse) were intraperitoneally injected into relapsing-remitting EAE induced SJL/J mice. Capsaicin (5 μg/mouse) was subcutaneously injected into the whiskers or forefeet of EAE-recovered mice. Anti-CX3CL1 antibody (10 μg/mouse), atenolol (10 μg/mouse), and clodronate liposomes 300 (100 μg/mouse) (Katayama Chemical Industries, Osaka) were infused into the subarachnoid CSF from the cisterna magna (*Xie et al., 2013*). 6-OHDA (5 mg/mouse) was twice intraperitoneally injected into EAE-recovered mice before ligation (*Seeley et al., 2013*).

## Brain microinjection

The head of an anesthetized mouse was placed in a stereotaxic device. Fur above the skull was shaved, and the skin was cleaned with 70% ethanol. A 30-gauge needle was lowered toward the ACC (AP 0.7 mm; ML 0.3 mm; VD 1.75 mm), and MK801 or L-Homocysteic acid (3 μg/μl and 100 mM, respectively, 0.5 μl each delivered over 90 s) were injected as described previously (*Kim et al., 2011*).

## Blood Flow analysis

Blood flow volume was measured in blood vessels from each tissue in C57BL/6 mice (6–8 weeks old) in the presence or absence of ligation using Omegazone OZ-1 (Omegawave, Tokyo).

## Antigen presentation assay

Naïve CD4 T cells from 2D2 mice and CD11b$^+$ cells from EAE-recovered mice were sorted using anti-CD4 microbeads and anti-CD11b microbeads, respectively (Miltenyi Biotec). The resulting CD4$^+$ T cell-enriched population ($1 \times 10^5$ cells) was cocultured with the isolated CD11b$^+$ cells ($5 \times 10^4$ cells) without MOG-peptide addition in a 96 well plate for 3 days. IL-2 levels in cell culture supernatants were determined using ELISA kits (eBioscience).

## Parabiosis

Parabiosis was performed as described previously (*Duyverman et al., 2012*). In brief, the right side of C57BL/6 (CD45.2+) mice and the left side of C57BL6-SJL (CD45.1+) mice were shaved and sterilized. Matching skin incisions were made from the base of the foreleg to the base of the hind leg of each mouse. The skin flaps of the parabiotic pair were attached by 6-0 polyglycolic acid suture, and the skin was closed with clips to finalize the parabiosis surgery. After 2 weeks, pathogenic CD4 T cells from EAE induced CD57BL/6 (CD45.2+) mice were transferred to each mouse. The infiltrated monocyte/macrophage was analyzed after mice recovered from the EAE symptoms.

## Immobilization stress

EAE-recovered mice were subjected to immobilization stress in a plastic tube for 30 min/day over 7 days (*Yoshihara and Yawaka, 2013*).

## Forced swimming model

EAE-recovered mice were subjected to the forced swim model in a tub for 15 min/day over 4 days as described previously (*Stone and Lin, 2011*).

## Statistical analysis

Student's t tests (two-tailed) and ANOVA tests were used for the statistical analysis of differences between two groups and that of differences between more than two groups, respectively.

## Study approval

All animal experiments were performed following the guidelines of the Institutional Animal Care and Use Committees of the Institute for Genetic Medicine, the Graduate School of Medicine, Hokkaido University (Sapporo, Japan) and the Graduate School of Frontier Bioscience and Graduate School of Medicine, Osaka University (Osaka, Japan).

## Acknowledgements

We appreciate the excellent technical assistances provided by Dr Jiang, and Ms Kumai, and thank Ms R Masuda for her excellent assistance. We thank Dr P Karagiannis (CiRA, Kyoto University, Kyoto, Japan) for carefully reading the manuscript. We appreciate Dr Tadamitsu Kishimoto (Osaka University, Osaka, Japan) for providing the anti-mIL-6 receptor antibody and Dr Y Nakatsuji (Osaka University) for important discussions about MS patients. We also appreciate Dr Y Okuyama (Osaka University) and Dr Y Ohira (Osaka University) for important discussions and carefully reading the manuscript. This work was supported by KAKENHI (DK, YA, TA, MM, and TH), Takeda Science Foundation (MM), Institute for Fermentation Osaka (MM), Mitsubishi Foundation (MM), Mochida Memorial Foundation for Medical and Pharmaceutical Research (DK), Suzuken Memorial Foundation (YA), Japan Prize Foundation (YA), Ono Medical Research Foundation (YA), Kanzawa Medical Research Foundation (YA), Kishimoto Foundation (YA), Nagao Takeshi Research Foundation (YA), Tokyo Medical Research Foundation (MM and YA), JSPS Postdoctoral Fellowship for Foreign Researchers (AS), the JST-CREST program (MM and TH) and the Osaka Foundation for the Promotion of Clinical Immunology (MM).

## Additional information

### Funding

| Funder | Grant reference | Author |
| --- | --- | --- |
| Takeda Science Foundation | | Yasunobu Arima, Masaaki Murakami |
| Naito Foundation | | Masaaki Murakami |
| KAKENHI | | Yasunobu Arima, Daisuke Kamimura, Toru Atsumi, Toshio Hirano, Masaaki Murakami |
| Institute for Fermentation, Osaka | | Masaaki Murakami |
| Mitsubishi Foundation | | Masaaki Murakami |
| Mochida Memorial Foundation for Medical and Pharmaceutical Research | | Daisuke Kamimura |
| Suzuken Memorial Foundation | | Yasunobu Arima |
| Japan Prize Foundation | | Yasunobu Arima |
| Ono Medical Research Foundation | | Yasunobu Arima |
| Kanzawa Medical Research Foundation | | Yasunobu Arima |
| Kishimoto Foundation | | Yasunobu Arima |
| Nagao Takeshi Research Foundation | | Yasunobu Arima |
| Tokyo Medical Research Foundation | | Yasunobu Arima, Masaaki Murakami |
| Japan Society for the Promotion of Science | Postdoctoral Fellowship for Foreign Researchers | Andrea Stofkova |
| JST-CREST program | | Toshio Hirano, Masaaki Murakami |
| Osaka Foundation for the Promotion of Clinical Immunology | | Masaaki Murakami |

The funders had no role in study design, data collection and interpretation, or the decision to submit the work for publication.

### Author contributions

YA, DK, Acquisition of data, Analysis and interpretation of data, Drafting or revising the article; TA, MH, NN, AS, TO, KH, YM, YY, Acquisition of data, Analysis and interpretation of data; TK, YO, IK, Analysis and interpretation of data, Contributed unpublished essential data or reagents; YM, SS, Conception and design, Drafting or revising the article; PW, Drafting or revising the article, Contributed unpublished essential data or reagents; SS, MP, Analysis and interpretation of data, Drafting or revising the article, Contributed unpublished essential data or reagents; TY, Conception and design, Analysis and interpretation of data; TH, Conception and design, Analysis and interpretation of data, Drafting or revising the article; MM, Conception and design, Acquisition of data, Analysis and interpretation of data, Drafting or revising the article, Contributed unpublished essential data or reagents

### Ethics

Animal experimentation: All animal experiments were performed following the guidelines of the Institutional Animal Care and Use Committees of the Institute for Genetic Medicine, the Graduate School of Medicine, Hokkaido University and the Graduate School of Frontier Bioscience and Graduate School of Medicine, Osaka University with protocol numbers 2014-0083 and 2014-0026.

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
