## [Decision Letter]

Thank you for submitting your work entitled “A pain-mediated neural signal induces relapse in multiple sclerosis models” for peer review at *eLife*. Your submission has been favorably evaluated by Tadatsugu Taniguchi (Senior editor) and three reviewers, one of whom is a member of our Board of Reviewing Editors.

The reviewers have discussed the reviews with one another and the Reviewing editor has drafted this decision to help you prepare a revised submission.

Summary:

This is an interesting manuscript with an intriguing plot. Its specific novelty is a connection between pain perception via activation of TRPV1 sensory neurons to activation in the anterior cingulate cortex on the one hand, and a local epinephrine/norepinephrine-dependent induction of CX3CL1/CX3CR1 in the L5 spinal cord on the other. Findings downstream of that event in the L5 region, leading to accumulation of macrophages, CD4 and CD8 T cells, providing inflammatory cytokines and causing a paralytic relapse, then follow expectations from the authors' earlier work (Arima et al., Cell, 2012).

Essential revisions:

While this is a very interesting demonstration of connectivity between pain perception and inflammation modulation, it would be useful to have some major concerns addressed.

1) Some of the immunohistochemistry shows fairly minimal staining with a high background. For example changes in MHC class II and CD4 expression are not convincing, making quantification uncertain. It would be useful to have cell counts and cell-surface protein expression profiles of the cells at L5 quantified as in Figure 4 for the major points being made.

2) The temporal aspects of the four-step sequential model proposed need clarification. In the Discussion, the 4-step model is laid out as sequential steps in a process. However, the present data (Figure 4) are consistent all of the cell types, being recruited at a similar rates and not in distinct waves. It is possible that there are positive feedback loops that produce this effect, or that all of the cell types are recruit due to a similar set of chemokines/cytokines. These issues (as for example in Figure 5) need to be clarified further, and with robust methodology as in Figure 4.

3) It would be useful to have some explanation of how pain perception in any anatomical locality leads, via the anterior cingulate cortex, to local epinephrine/norepinephrine-dependent induction of CX3CL1/CX3CR1 specifically in the L5 spinal cord, which cannot be mimicked by systemic epinephrine/norepinephrine.

4) Some explanation of how pain-induced epinephrine/norepinephrine leads to activation of a CX3CL1-CX3CR1 axis leading to inflammation, while soleus sensory input-induced epinephrine/norepinephrine leads to activation of an IL6-CCL20-CCR6 axis (Arima et al., Cell, 2012), both in L5, would be very helpful.

5) The caveat that clinically there does not seem to be a direct correlation between pain intensity and MS disability or lesional load should be noted, as also the lack of natural relapses in the C57BL/6 mouse strain.

---

## [Author Response]

*1) Some of the immunohistochemistry shows fairly minimal staining with a high background. For example changes in MHC class II and CD4 expression are not convincing, making quantification uncertain*.

We have incorporated more convincing immunohistochemistry results in Figures 4, 5 and 6.

*It would be useful to have cell counts and cell-surface protein expression profiles of the cells at L5 quantified as in*
Figure 4
*for the major points being made*.

We have added the number of MHC class II+ cells in the dotted circles in the Figures. These values were determined by using serial sections with Hoechst nucleus staining. The ratio of MFI to cell number of the accumulated cells around the L5 ventral vessels before and after pain induction were similar. Indeed, the number of MHC class II+CD11b+ cells in each spinal cord, including L5, were basically unchanged before and after overnight pain induction (8 hours). However these cells accumulated around the two ventral blood vessels after pain induction. Therefore, it was difficult to evaluate their accumulation around the vessels by FACS. These data are shown in Figures 4, 5 and 6, Figure 1—figure supplement 2, Figure 2—figure supplement 1, and Figure 2—figure supplement 2 and described in the third paragraph of subsection “A pain-mediated neural pathway accumulates MHC class II+CD11b+ cells at L5 ventral vessels via sympathetic-mediated chemokine expression in EAE-recovered mice” and in their Figure legends.

*2) The temporal aspects of the four-step sequential model proposed need clarification. In the Discussion, the 4-step model is laid out as sequential steps in a process. However, the present data (*Figure 4*) are consistent all of the cell types, being recruited at a similar rates and not in distinct waves. It is possible that there are positive feedback loops that produce this effect, or that all of the cell types are recruit due to a similar set of chemokines/cytokines. These issues (as for example in*
Figure 5*) need to be clarified further, and with robust methodology as in*
Figure 4.

As suggested by the reviewers, the flow cytometry data in Figure 4 showed that all cell types were recruited at similar rates and did not accumulate in distinct waves. It should be pointed out that the sensitivity of the FACS analysis is low with regards to the MHC class II+CD11b+ cells and CD4+ T cells found in the immunohistochemistry. Therefore, FACS analysis will not reveal sequential steps. The results of the blocking experiments, however, do show four steps: Regarding the 1^st^ step, blockades of sensory pathways by using A803467, TRPV1-deficient hosts, and MK801 at the ACC region suppressed somatosensory activations, sympathetic activations, and MHC class II+CD11b+ cell accumulation in the ventral vessels, and EAE-relapse. Regarding the 2^nd^ step, we used atenolol and 6-OHDA to suppress sympathetic pathway and MHC class II+CD11b+ cell accumulation in the ventral vessels and/or EAE-relapse. However, this treatment did not affect sensory-mediated somatosensory activations, suggesting that sympathetic activation is a downstream event after sensory activation. Also regarding the 2^nd^ step, blockade of CX3CL1 did suppress MHC class II+CD11b+ cell accumulation in the ventral vessels and EAE-relapse but not somatosensory activations and sympathetic activations, suggesting that CX3CL1 expression is a downstream event after sensory and sympathetic activation. Regarding the 3^rd^ step, anti-CD4 antibody application suppressed CD4+ T cell accumulation and EAE-relapse but not somatosensory activations, sympathetic activations, or MHC class II+CD11b+ cell accumulation in the ventral vessels, suggesting that CD4+ T cell accumulation is a downstream event of sensory and sympathetic activation and of MHC class II+CD11b+ cell accumulation in the ventral vessels. MHC class II+CD11b+ cell accumulation in the ventral vessels as observed within 8 hours (overnight). Regarding the 4^th^ step, blockades of IL-17 and IL-6 signal by neutralizing antibodies suppressed EAE-relapse but not somatosensory activations, sympathetic activations, MHC class II+CD11b+ cell accumulation in the ventral vessels, or CD4+ T cell accumulation, suggesting that the inflammation amplifier activation by cytokines including IL-17 and IL-6 is a downstream event after sensory and sympathetic activation, MHC class II+CD11b+ cell accumulation in the ventral vessels, and pathogenic Th17 cell accumulation in the ventral vessels. These results clearly suggested there are the four-steps that occur sequentially and are critical for the development of pain-mediated EAE relapse in our EAE models. We described these matters in the sixth paragraph of the Discussion and a detailed methodology in Figure 4.

*3) It would be useful to have some explanation of how pain perception in any anatomical locality leads, via the anterior cingulate cortex, to local epinephrine/norepinephrine-dependent induction of CX3CL1/CX3CR1 specifically in the L5 spinal cord, which cannot be mimicked by systemic epinephrine/norepinephrine*.

We hypothesized pain-mediated sympathetic signaling acts via a high concentration of norepinephrine at the sympathetic neurovascular connection, but not through the systemic induction of hormones such as epinephrine and norepinephrine, which play a major role in the development of EAE relapse, because neither an accumulation of MHC class II+CD11b+ cells at L5 ventral vessels nor EAE relapse in remittent mice that had immobilization stress or forced swimming were observed, even though serum norepinephrine and epinephrine increased just like in the pain-induction condition, which did induce MHC class II+CD11b+ cell accumulation as well as EAE relapse. However, epinephrine and norepinephrine induced CX3CL1 expression dose-dependently in MHC class II+CD11b+ cells, which are also present in spleen and lymphonodes, and endothelial cells and fibroblasts induced various chemokines, including CCL20, after stimulation by epinephrine and norepinephrine, particularly in the presence of NFkB and STAT3 activation (1; 2). Therefore, we hypothesized that systemic epinephrine/norepinephrine molecules induced a systemic pro-inflammatory state via chemokine expression. We described these matters in the eighth paragraph of the Discussion.

*4) Some explanation of how pain-induced epinephrine/norepinephrine leads to activation of a CX3CL1-CX3CR1 axis leading to inflammation, while soleus sensory input-induced epinephrine/norepinephrine leads to activation of an IL6-CCL20-CCR6 axis (Arima et al., Cell, 2012), both in L5, would be very helpful*.

Regarding the induction of different chemokine-mediated axes by different sympathetic neurons (pain vs. anti-gravity), we should point out that we used different hosts in these two cases. We used normal mice for anti-gravity experiments, but EAE-recovered mice for pain-mediated ones. In other words, the numbers of MHC class II+CD11b+ cells are completely different: normal mice have almost no MHC class II+CD11b+ cells in the CNS, while EAE-recovered mice have many MHC class II+CD11b+ cells in the CNS particularly in the L5 cord. In EAE-recovered mice, a regional norepinephrine output via pain inductions functions on MHC class II+CD11b+ cells as well as endothelial cells around the ventral vessels of the spinal cords, causing the expression of CX3CL1 and accumulation of MHC class II+CD11b+CX3CR1+ cells in a manner dependent on the CX3CL1-CX3CR1 axis. These accumulated MHC class II+CD11b+CX3CR1+ cells present autoantigens to pathogenic Th17 cells located in the blood stream. On the other hand, in normal mice, a regional norepinephrine output via anti-gravity responses functions just on endothelial cells because of there are almost no MHC class II+CD11b+ cells in the CNS. This case results in CCL20 expression, which causes the accumulation pathogenic Th17 cells that express the CCR6 receptor, IL-17 and IL-6. We have described these matters in the Discussion.

*5) The caveat that clinically there does not seem to be a direct correlation between pain intensity and MS disability or lesional load should be noted, as also the lack of natural relapses in the C57BL/6 mouse strain*.

As suggested, we note these matters in the revised text in Results:

“Consistent with these results, it is known that the lack of natural relapses in the C57BL/6 mouse strain”, and Discussion:

“Thus, there does not seem to be a direct correlation clinically between pain intensity and MS disability or lesional load.”